# South Korean Nurses’ Experiences with Patient Care at a COVID-19-Designated Hospital: Growth after the Frontline Battle against an Infectious Disease Pandemic

**DOI:** 10.3390/ijerph17239015

**Published:** 2020-12-03

**Authors:** Nayoon Lee, Hyun-Ju Lee

**Affiliations:** College of Nursing, Catholic University of Pusan, Busan 46252, Korea; nayoon@cup.ac.kr

**Keywords:** COVID-19, pandemics, occupational stress, post-traumatic growth, nursing care, qualitative research

## Abstract

COVID-19 is a respiratory disease caused by a novel coronavirus that quickly spread worldwide, resulting in a global pandemic. Healthcare professionals coming into close contact with COVID-19 patients experience mental health issues, including stress, depression, anxiety, post-traumatic stress disorder, and burnout. This study aimed to explore the experiences of COVID-19-designated hospital nurses in South Korea who provided care for patients based on their lived experiences. Eighteen nurses working in a COVID-19-designated hospital completed in-depth individual telephone interviews between July and September 2020, and the data were analyzed using Giorgi’s phenomenological methodology. The essential structure of the phenomenon was growth after the frontline battle against an infectious disease pandemic. Nine themes were identified: Pushed onto the Battlefield Without Any Preparation, Struggling on the Frontline, Altered Daily Life, Low Morale, Unexpectedly Long War, Ambivalence Toward Patients, Forces that Keep Me Going, Giving Meaning to My Work, and Taking Another Step in One’s Growth. The nurses who cared for patients with COVID-19 had both negative and positive experiences, including post-traumatic growth. These findings could be used as basic data for establishing hospital systems and policies to support frontline nurses coping with infectious disease control to increase their adaption and positive experiences.

## 1. Introduction

COVID-19 is a respiratory disease caused by a novel coronavirus that was identified in 2019. Since the first reported case in December 2019, it has spread worldwide, leading to the declaration of a global pandemic, which is the highest infectious disease alert level [1]. As of 23 November 2020, there have been 58,425,681 confirmed cases and 1,385,218 deaths associated with COVID-19 worldwide [2], representing a significantly higher number of confirmed cases and deaths compared to previous pandemics such as the 1968 Hong Kong Flu and 2009 Swine Flu Pandemics [3]. In addition, different countries have implemented various measures such as travel restrictions, social distancing, and school closing to prevent the spread of the pandemic; thus, COVID-19 has had a greater impact on daily life than past infectious diseases [4].

Research has indicated that due to the ongoing COVID-19 pandemic, people are experiencing high levels of stress, depression, and anxiety, as well as levels of post-traumatic stress that are above the cut-off for post-traumatic stress disorder (PTSD) [5,6,7]. Healthcare professionals who come in close contact with COVID-19 patients are deeply affected by this novel virus and experience mental health challenges, including stress, depression, anxiety, PTSD, and burnout [8,9,10,11,12,13], which can have deleterious effects on their quality of life [11] and increase their turnover intention [14]. In addition, their concentration, comprehension, and decision-making ability can be affected by these psychological difficulties, which could, in turn, affect infectious disease management [15]. Therefore, developing strategies to reduce the negative impact of COVID-19 among healthcare professionals is necessary.

Nurses are the frontline healthcare professionals in the closest contact with infected patients and spend the most time with them, contributing to nurses facing greater difficulties than other groups of healthcare professionals, including doctors, during a global pandemic [9,16,17]. Controlling a global pandemic requires appropriate policies not only for infectious disease management, but also for nurses working on the frontline. Therefore, it is very important to understand the perspective of the nurses who care for COVID-19 patients on their frontline experiences during a global pandemic.

Previous qualitative studies have been conducted to provide an in-depth investigation of the effects of COVID-19 on nurses; however, this research has focused on positive coping strategies used by nurses to overcome fatigue and burnout due to excessive workload, stress, fear of infection, and the discomfort associated with wearing personal protective equipment (PPE) [18,19,20]. After COVID-19 was declared a pandemic, each country implemented its own prevention and control measures, and each faced different situations with varying infection and mortality rates. Therefore, the experiences of nurses who are caring for COVID-19 patients are expected to vary depending on the infection prevention and control strategies, hospital resources, and socio-cultural background of each country.

In South Korea, the first confirmed case of COVID-19 was reported on 20 January 2020, after which the infection spread rapidly [21]. Even during the early stages of the pandemic, the South Korean government attempted to block the infection spread by disclosing confirmed patients’ travel routes and providing free diagnostic tests for those in contact with confirmed patients or experiencing related symptoms. The South Korean government also enabled infected patients to receive low-cost treatment and implemented strict prevention and control policies, including the distribution of masks to make mask wearing a part of daily life. As a result, South Korea experienced fewer confirmed cases and a lower mortality rate than other countries [22]. However, despite the extensive prevention and control policies implemented by the South Korean government and the protracted length of the COVID-19 pandemic, studies on identifying the overall experiences of nurses providing care for patients with COVID-19 are lacking.

Accordingly, we applied the descriptive phenomenology methodology by Giorgi [23] to provide an in-depth exploration of the meaning and nature of the patient care experiences of nurses at a COVID-19-designated hospital in South Korea using the lived experiences of the participants. Our objective was to use our findings to promote a broad understanding of the nurses’ COVID-19 patient care experiences and provide basic data to develop support programs for nurses facing an unavoidable pandemic so that the pandemic’s negative impacts on the nurses can be minimized and the nurses can effectively cope with the situation.

## 2. Materials and Methods

### 2.1. Study Design

This study was conducted using the steps of Giorgi’s phenomenological method, which are presented in Figure 1. Giorgi’s descriptive phenomenological study was conducted to explore the lived experiences and essential structure of the experiences of a group of nurses at a COVID-19-designated hospital who were providing patient care. Giorgi’s method [23] focuses on revealing the vivid meaning of the participants’ experience; describes each participant’s unique experience in detail; and incorporates the participant’s entire experience according to changes in context, relationships, time, and perspectives to derive a general structure.

### 2.2. Participants and Setting

Among nurses who work in a government-designated COVID-19 hospital in Busan, South Korea, our study selected those having at least one year of patient care experience and at least two months of work experience in a COVID-19 isolation ward. Nurse managers who worked in the isolation ward but were not directly involved in patient care were excluded. The first participant, who was willing to share her experience, was recommended by nurses in the hospital, after which snowball sampling was applied to obtain recommendations for additional participants. The study population included 18 participants who were all women. Eight, eight, and two of the participants were in the 20–29, 30–39, and 40–49 years age groups, respectively. Four participants were married, while the remainder were single. The participants’ mean COVID-19 patient care experience was 5.33 ± 2.22 months, and their mean total work experience was 7.44 ± 5.90 years, with a range between 2 and 22 years. Last, three participants had previous Middle East respiratory syndrome (MERS) patient care experience.

### 2.3. Data Collection and Procedure

Data were collected using in-depth interviews with each individual from 8 June to 25 September 2020. Given the ongoing COVID-19 pandemic, all interviews were conducted over the telephone (mobile phone). The first interview with each participant lasted approximately 60 min. Additional interviews, lasting approximately 10–20 min each, were conducted over the telephone with two participants, due to the need for reconfirmation of responses with unclear meaning. Data collection was completed when it was determined that no new information about nurses’ work experiences as a COVID-19-designated hospital nurse could be obtained. The in-depth interviews began with an unstructured question, “What can you tell us about your experience of providing patient care as a COVID-19-designated hospital nurse?” As the interviews progressed, the researchers asked the following additional questions to derive a variety of concepts and categories. (1) How did you feel about experiences with patient care at a COVID-19–designated hospital? (2) What can you tell us about your challenges while caring for COVID-19 patients? (3) How did you get through the hard times while caring for COVID-19 patients? (4) Can you tell us about any good experiences that happened while caring for COVID-19 patients?

During the interviews, a conscious effort was made to listen to the participant while avoiding conversation and leading questions that could inadvertently influence the direction of the responses. The recorded interviews were played back and transcribed within three days of the interview by the researchers to assure that the vivid information and feelings were preserved. The researchers who conducted the interviews increased the accuracy of data by including notes that had been written about non-verbal expressions during the interviews, including pauses in between conversations, the pace of speech, and expression of emotions.

### 2.4. Data Analysis

Data analysis was performed following the analytical procedures for descriptive phenomenology outlined by Giorgi [23] that are presented in Figure 1. Meaning units were separated from the collected data during the coding process, and the qualitative data analysis program NVivo Pro 12.0 was used to facilitate efficient structuring of the data. A specific analytical procedure was followed. First, the complete statements from all 18 participants were read repeatedly from start to finish to ascertain the overall feeling of the data. The goal at this stage is to identify the general meaning behind the overall data, which becomes the basis for analysis in the next stage [23]. After repeatedly listening to the recording and reading the manuscript, researchers deliberate the meanings implied by the statements with a phenomenological attitude in order to understand the meanings expressing the experiences of the participants from their perspectives while maintaining an objective attitude as much as possible.

The second step involves distinguishing the meaning units from the point of view of phenomenological psychology by focusing on the phenomenon covered in the research topic. We imported the transcribed verbatim interview into NVivo Pro 12.0, after which the statements were reread. Those statements considered important in representing the work experiences of COVID-19-designated hospital nurses, despite different styles of expression and vocabulary used, were separated as meaning units. In this process, around 350 meaningful statements related to the experiences of the participants were extracted.

In the next step, while reviewing the meaning units that were derived for theme refinement, the participants’ expressed experiences were converted into academic descriptions. The third step is the heart of the method [23]. The researchers replaced overlapping meaning units with a single meaning unit and then confirmed their results. By organizing these units into a table, we were able to contrast each meaning unit to the others and associated them with an overall meaning, thereby transforming the units into words considered to be the most suitable. At this time, as proposed by Giorgi [23], words and idiomatic expressions that were sufficiently appropriate in the ordinary world were used, and in cases in which an academic term was absent, the meaning units were organized using the terms expressed by the participant.

The fourth step involved a process of determining the essence of the units by defining and integrating the essential structure from the phenomenon based on the transformed meaning units. The researchers shared each meaning unit in advance and held three meetings thereafter to exchange, integrate, and modify the opinions for classification and categorization into the themes. Subsequently, we conducted a review to determine whether the themes appeared within each participant’s statements and were combined to describe the essential structure.

### 2.5. Rigor

The quality of this study was assured using the evaluative criteria for trustworthiness developed by Lincoln and Guba [24]. For credibility, the researchers listened to the participants’ statements while minimizing any bias or prejudice during the interview. The recorded data were played back and transcribed verbatim within three days of the interview to ensure that the vivid details and feelings were preserved. The analyzed meaning units were shown to the participants for review to ensure that they were presented as intended. For the transferability of the derived results, the findings were presented to three COVID-19-designated hospital nurses who did not participate in the study to confirm their agreement. For dependability, we tried to analyze the data based on Giorgi’s method [23], and any differences in opinion were discussed and modified to ensure consistent results between the researchers. For confirmability, the researchers listened to the participants’ statements while minimizing any bias or prejudice during the interview, and we attempted to view the data with a fresh set of eyes by suspending judgment and deriving the meanings about the experiences of the participants.

### 2.6. Ethical Considerations

This study was approved by the institutional board of Catholic University of Pusan (Approval No. CUPIRB-2020-007). Prior to data collection, all participants were informed of the objectives, interview methods, and data storage of this study. We also explained confidentiality, discontinuation of participation, and withdrawal of consent before obtaining consent for participation in the study and recording the interviews. Each participant received a small token of appreciation after completing the interviews.

## 3. Results

The study findings showed that nine themes and 31 subthemes emerged from 178 meaning units by the phenomenological analysis (Table 1). The essential structure of the lived patient care experiences of nurses in a COVID-19-dedicated hospital was identified as growth through struggles on the frontline against a global pandemic, showing how their experiences changed over time based on their situational context.

As the novel infectious disease outbreaks began, COVID-19 hospital nurses were pushed into the forefront of the pandemic without any preparation because they were nurses working in a public hospital and were struggling against an invisible enemy while standing on the frontline. During this process, they experienced many changes at work and home due to COVID-19, and their motivation decreased as their efforts were not properly recognized. They grew tired of the protracted pandemic and experienced various negative emotions while feeling ambivalent toward the patients who were wholly dependent on them. What allowed them to continue their work was social support, such as encouragement from peers, family, friends, patients, and the public. As the participants went through this process, they developed a positive meaning from their work and saw themselves growing in their role as dedicated COVID-19 nurses.

### 3.1. Pushed into the Battlefield without Any Preparation

#### 3.1.1. Unavoidable Duty

The participants were required to begin caring for COVID-19 patients, regardless of their personal inclinations, once their workplace was designated as a COVID-19 hospital at the beginning of the COVID-19 pandemic.


*Because our hospital is a public hospital, I suspected that we would care for infected patients once an infectious disease outbreak occurs. However, when the mayor announced that our hospital would be officially designated as a COVID-19 hospital, I got scared since I had never done it before.*

*Feeling of being sent to a battlefield? Feeling a sense of unavoidable duty since the country was facing a major crisis.*


#### 3.1.2. Limited Preparation Time

As the number of confirmed COVID-19 cases increased, South Korea initiated urgent preparation measures, including transferring existing inpatients to other hospitals and expanding the negative pressure and isolation units to care for COVID-19 patients following community outbreaks. As all the pandemic situations were urgent, participants were not allowed time to gain sufficient training for caring for patients with infectious diseases. This situation made the participants nervous and less confident in their work.


*It was like a dream to see all that happen in a few days. I took a few days off and went to work, and I was immediately placed in a situation caring for COVID-19 patients.*

*Once I saw a senior nurse wearing PPE and I immediately put on PPE, looking at myself in the mirror. Then I went straight into the isolation u*
*nit. Everything was so urgent that I was embarrassed and anxious. I even doubted myself if I was wearing the PPE properly*


#### 3.1.3. Fear of an Unknown Enemy

During the early stage of the pandemic, most participants experienced fear because they were caring for patients with suspected or confirmed cases of COVID-19 while only wearing unfamiliar PPE and having received no clear information about the route of transmission, incubation period, and related symptoms of the novel virus.


*It was really scary before entering the isolation unit for the first time while wearing PPE. Even though I had a mask and goggles on, will those protect me… Is it safe to breathe…*

*There was no treatment, and I didn’t even know the route of transmission at first… I was so scared that it felt like entering hell.*


### 3.2. Struggling on the Frontline

#### 3.2.1. Constantly Changing Guidelines That Lack Details

The participants assigned to the isolation unit for confirmed patients during the early stage of the pandemic experienced much confusion due to the clinical guidelines not being very clear and constantly changing.


*The guidelines kept changing, and that was somewhat confusing. Even when the infection control office told us the guidelines, there were many ambiguities as we worked in the field, so we kept asking questions…*

*The guidelines kept changing. It made us anxious, and we kept thinking, “Is this right?” even though we were following the guidelines.*


#### 3.2.2. Discomfort Due to Personal Protective Equipment

The participants were required to wear PPE when entering isolation units to prevent infection. However, the process of donning and doffing PPE was unfamiliar and cumbersome, and the participants experienced many physical discomforts while working with PPE on. In addition, nursing practices were hindered due to communication problems, limited visibility, and limited sophisticated hand movements.


*First of all, just doing my work while wearing PPE was hot and difficult. When wearing an N95 mask, it was really difficult to breathe, and I even felt dizzy at times.*

*When I wore goggles, they would fog up, and I couldn’t see. And then, as it got foggier, drops of water would drip down inside the goggles. I could only do my work by looking through the drip mark. And I had to wear three layers of gloves and put in an IV line. My hands became ungainly, and it took much longer than usual. I felt like a beginner nurse with everything being new.*


#### 3.2.3. Unfamiliar Work

Due to the nature of the COVID-19 isolation unit, family members or assistants could not enter; therefore, participants were solely responsible for patient care. When they were first assigned to the isolation unit, the participants were not accustomed to this situation where they were responsible for everything, including feeding, changing diapers, changing positions, waste disposal, and hospital room maintenance, all while wearing PPE.


*Once the patient is admitted, the nurse has to take care of everything. Tidying up personal belongings and taking vital signs are basic stuff, but the nurses were responsible for everything, even feeding, putting away trays after eating, and cleaning the room. The number of trash cans that I personally had to take care of was over ten.*

*Because we had to be the caregiver, guardian, and nurse, I had to be hands-on for even the smallest things. That made it difficult. It’s not something I’m used to doing…*


#### 3.2.4. A Series of Unexpected Situations

The participants were unable to anticipate or prepare for the pandemic. This situation forced participants to work overtime as the number of confirmed cases increased, and they felt high levels of stress and helplessness.


*The number of confirmed cases, infection control instructions guidelines… Everything happened without warning. That’s what made us crazy. But even in that situation, we had to do our best. patients are rushed in all at once, we are completely out of our minds…*

*We reduced the number of nurses on duty since the number of patients decreased, but because several patients were admitted suddenly, I went to work in the early morning and got off work at night… Patients were admitted suddenly, and the charge nurse had to come to work on a scheduled off-day.*


### 3.3. Altered Daily Life

#### 3.3.1. Reinforced Infection Policies within the Hospital

Regarding the COVID-19 pandemic, the government issued social distancing guidelines. As a result, all employees, regardless of being a healthcare professional or not, were required to wear a mask and have their temperature checked when coming to and leaving work. Moreover, partitions were set up in the hospital cafeteria and people could not eat facing each other. The participants caring for COVID-19 patients were required to regularly be tested for COVID-19 regardless of whether they had any symptoms. In other words, participants were experiencing not only major changes in the patient care experience due to the COVID-19 pandemic but also changed situations in the workplace.

*We used to eat together with our masks off. After that* [a nurse became infected], *we stopped eating all together and were split up into teams to eat, separated from each other, and facing the wall. It’s necessary to prevent infection, but the great pleasure of talking over meals is gone.**After a colleague nurse was confirmed as having COVID-19, the hospital is testing* [healthcare professionals] *once a month for COVID-19. Tomorrow’s the test day. I’m always nervous until the results come back.*

#### 3.3.2. Becoming Sensitive to Even Minor Symptoms

Among the participants, some became suspicious even if their diagnostic test results were negative or felt anxious about becoming infected when unspecific symptoms, such as headache, nasal stuffiness, and sneezing, appeared.

*I had a severe cold. I didn’t have any fever at first, but my throat was hurting so much. I was anxious. Was it because the shield had ripped* [while caring for infected patients]*? I kept thinking about it. I kept coughing the next day. I was convinced that I became infected and spent the whole day crying.*
*I had a headache and mild fever. Did I get COVID-19? I was worried, and since I have a child at home, I went to my mom’s house and slept there.*


#### 3.3.3. Voluntarily Restricting Social Activities

Most participants refrained from leisure activities or avoiding gatherings with friends or acquaintances due to potentially being a source of viral transmission. For the same reason, they socially distanced from their closest family members, avoiding contact with their children or using a separate room at home.


*Because I need to be careful… I used to work out, but I don’t go to the gym any more… I used to enjoy getting together with people, but I can’t do it anymore, which makes me realize how precious my daily life was.*

*When I go home now, I wash my hands right away. I touch my child, but no kisses…*


#### 3.3.4. Cohort Isolation after the Confirmed Diagnosis of a Colleague

A nurse colleague from the hospital where the participants worked was diagnosed with COVID-19. Some participants were classified as having close contact with the colleague. They were not allowed to go home and were placed in cohort isolation for two weeks at the hospital.


*After hearing the news about a nurse being confirmed as having COVID-19, it made me think that COVID-19 isn’t just someone else’s concern.*

*About 90 people who worked together or came in contact with the nurse diagnosed with COVID-19 were classified as the “red group” and were placed in isolation at the hospital for 14 days.*


### 3.4. Low Morale

#### 3.4.1. My Labor Is Not Being Properly Recognized

Despite their struggles on the frontline of the pandemic, the participants experienced delays in direct financial compensation, such as bonuses, and did not receive indirect compensation, such as vacation time and benefits. Moreover, they also expressed dissatisfaction with spending relatively more time in direct patient care or labor-intensive work than the doctors despite both being healthcare professionals.

*Doctors and clinical pathologists do not go into the isolation unit often, and they come right back out once their work is done. Nurses, on the other hand, perform not only their nursing work but also have to assist those people* [doctors and clinical pathologists] *when they come in and tidy up after they leave… So, nurses spend much more time in the isolation unit and have more work to do.*
*I haven’t received any bonuses yet. I’ve heard many rumors about bonuses, some saying that doctors are going to get more bonuses. Nurses have the toughest time, but when I hear rumors like that, it is really disheartening and frustrating.*


#### 3.4.2. Being Treated Like a Virus

The participants shared their experiences of feeling like they had become people whom others avoided because they were nurses at a COVID-19-designated hospital, which led to discrimination in their daily lives. In addition, the participants’ families had a similar experience even though they were not infected. The stigma broke the hearts of the participants, and made them hide the fact that they were nurses at a COVID-19-designated hospital and restricted their own radius of action.


*I can’t tell others about working at this hospital because when I asked the taxi driver to take me to the hospital, he asked me if I worked there, and then he told me to get out. They don’t even deliver food to the hospital. I wasn’t infected with COVID-19, and I didn’t do anything wrong, but I had to stay at home. Because people don’t want contact with me.*

*My child goes to preschool. I told the teacher at preschool that I won’t be sending my child there for a while, just in case my child might get infected because of me. The teacher seemed relieved and happy when I mentioned that.*


#### 3.4.3. Strict Social Standards

The participants felt burdened and even angry at times due to the unusually strict social standards directed only at nurses. In particular, they mentioned that they thought about quitting after seeing critical posts flooding social networks and the attention received by a nurse colleague nurse diagnosed with COVID-19.


*When a nurse was diagnosed with COVID-19, it was from taking care of patients. However, she became a target of criticism, questioning whether she took off her PPE properly. Seeing that made me really angry. If I were put in that position, I would quit….*

*When I read the news, there were comments about why do healthcare professionals who deal with confirmed patients go out to eat or go to the gym… Honestly, we can go out to eat. We really get hurt when we see malicious comments like that.*


### 3.5. Unexpectedly Long War

#### 3.5.1. Despair with No End in Sight

After a cluster outbreak among members of a religious cult during the early stage of the pandemic, the number of confirmed cases gradually decreased owing to the prevention and control measures implemented by the government and social distancing practiced by the public. However, as the number of confirmed cases rose again due to a nightclub cluster outbreak, mass gatherings for religious services, and anti-government protests, the participants worked day-to-day while worrying that these situations would continue to repeat.


*The number of confirmed cases has been on the rise again as people have gone to nightclubs, anti-government protests, and religious gatherings without following the government’s guidelines for infection control. They don’t even have a screening test for fear of being criticized when confirmed, and they infect people around them. Being anxious about no end in sight. Will this ever end? Everyone is so tired and crazed. Nobody thought this would last this long…*


#### 3.5.2. Tired Body and Mind

As stays in the isolation units became longer due to workforce shortages and the continuation of the pandemic, participants began experiencing headaches, dizziness, and fatigue, together with physical decline. They also expressed symptoms of depression due to the prolonged pandemic.


*How long do I have to work like this?… They call it the corona blues. I think that applies to me.*

*In the beginning, we had the personnel to take shifts, so the time spent in the isolation unit was short. Back then, I felt refreshed after showering, but that’s not the case now. I’m taking a lot of analgesics due to headaches.*


#### 3.5.3. Concerns about General Nursing Competency

Most of the participants were nurses who worked in the general ward before being assigned to the isolation ward. As most patients admitted to the isolation unit are in relatively good condition, and the work the nurses perform involves mostly basic nursing that does not require a high level of skill, they expressed concerned about being able to perform their work properly when they return to their ward after the pandemic ends. Such concerns tended to be greater among nurses with less work experience.


*In my two years of nursing experience, I spent seven months exclusively caring for COVID-19 patients. It’s because when a COVID-19 patient becomes seriously ill, that patient is transferred to another hospital. Actually, nursing while wearing PPE involves just the basics… I’m gaining experience, but…*

*Would I be able to do my work properly when I go back to the ward I was working in? I think I will be confused from not working there for so long.*


### 3.6. Ambivalence toward Patients

#### 3.6.1. Having a Bias

The participants stated that when they encounter patients admitted owing to a cluster outbreak from attending a church service or gathering, they lacked sympathy and did not want to understand the patients’ situation because they had not followed the basic prevention and control rules.


*In the beginning, there were many patients who were members of a religious cult. Maybe that’s the reason for thinking, “They’re somewhat weird”…*

*The government emphasized no mass gatherings, and anyone who’s been to a mass gathering should get tested… They didn’t cooperate… Because that is how I viewed them, I couldn’t really sympathize with them like other patients…*


#### 3.6.2. Becoming Angry at Uncooperative Attitudes

Due to the nature of the isolation unit, the patients could not leave or have visitors. The participants complained about their difficulty hiding their rising anger from having patients ask the nurses to bring delivered goods or food to them or cause a ruckus by trying to leave the isolation unit against the infection prevention guidelines.


*The patients wanted to eat outside food, so they had food delivered to the hospital lobby and asked us to go get it for them… We told them not to, but they kept doing it…*

*When a patient requests something, it takes time for us to put on PPE, so we can’t go in right away. Some patients become angry that we came a bit late, and one even threw a blood pressure meter at us.*


#### 3.6.3. Feelings of Pity

On the other hand, the participants heard various stories from patients about how they became infected and how their lives had been upended due to hospitalization. When hearing this, the participants would feel bad thinking that something similar could happen to themselves or their family.


*Among the isolated patients, there were some who could not accept the confirmed diagnosis and cried about needing to go home. When I saw that, I thought about how that could happen to me or to my mom or family.*

*After being admitted to the isolation unit, [they were] just sitting there crying without eating, crying while talking to family…There are many patients like that. It makes me sad.*


#### 3.6.4. Solidarity with the Patients

As the participants spent more time with the isolation patients, they found themselves becoming increasingly empathetic to the patients. When a patient for whom they were caring was discharged after their case resolved, the participants became elated and even cried at the thought of overcoming such a dangerous situation together.


*When the test result came back negative for the first time after being admitted, I was so happy and cried together with the patients.*

*There is something different than regular patients because of the thought that we provided care under the dangers of an unknown infectious disease and overcame it together.*


### 3.7. Forces That Keep Me Going

#### 3.7.1. Concerns from Family and Friends

Family members and close friends were worried and concerned about the participants having to care for COVID-19 patients, especially considering there was no existing treatment. When they could not meet face-to-face, family and friends often greeted the participants by phone or through social network services.


*My family worries about me a lot and calls me often to take care of my health.*

*My friends know I work in a COVID-19-designated hospital. They always call me to cheer me up and tell me that I’m really cool.*


#### 3.7.2. Patients Showing Their Appreciation

The participants stated that when patients told them to “keep up the good work” despite their own psychological burdens of being confined to the isolation unit until complete recovery or patients expressed “thank you very much” after being discharged, these expressions became a source of strength.


*The patient said, “If I didn’t get infected, you wouldn’t have to do this… So much trouble for you.”*
*Unlike regular patients, they* [COVID-19 patients] *have been pushed to the brink from an extreme situation. I think that’s why they lean on us and depend on us. So, they express their gratitude a lot. Even after being discharged, they sent text messages saying they will not forget how grateful they were during their stay.*

#### 3.7.3. Public Support

The “Thank You Challenge” was a public participation campaign that was started for medical professionals working during the COVID-19 pandemic, along with relief supplies sent from various groups, letters written by kindergarteners, and messages of encouragement from the public. These all served as motivation for the participants to work even harder.


*I was so grateful that people gave recognition to our struggles through the “Thank You Challenge.” Not only food, but since we have to wash frequently, people have given shampoo and cosmetics to us. It felt great, and I was proud whenever we received donated goods…*

*I got emotional when reading letters written by kindergarteners with messy writing that said, “Dear nurse… You are working really hard. Thank you very much for treating corona patients.”*


#### 3.7.4. Strengthened Camaraderie

In the initial stages of the COVID-19 pandemic, nurses from various wards began working together in the isolation unit to care for the infected patients. Their relationship was initially awkward, but it changed to showing consideration and encouragement for one another within a short period, as they were working closely together in a unique circumstance of caring for patients suffering from a novel infectious disease.


*Before, I was criticized for mistakes… Now, everything is new for everyone. Even if I am not proficient at something, they say it’s okay, just do it carefully… It’s an atmosphere of everyone encouraging each other. I feel it’s more so because we are overcoming a difficult situation together.*

*Before, patients or guardians would come by the nurses’ station, so that the nurses couldn’t talk about personal matters. Now, the patients are in the isolation unit, and we have time to talk amongst ourselves. So we talk to each other to relieve stress and become closer to each other.*


### 3.8. Giving Meaning to My Work

#### 3.8.1. Calling to Do the Work Expected of Me

The participants went on to care for COVID-19 patients, not by their own volition, but simply because they were nurses in a government-designated COVID-19 hospital. However, as time passed, they completed their work with the mindset that someone must do this work and that only they, themselves, could do it.


*Because it is a public institution, I always think that if something happens in the city, I’ll be put to work. Even if there is another infectious disease outbreak, I will do it again. It’s my job.*

*You can think of it as if I prevent the spread of the virus by treating these patients, then that is something I’m doing to protect my family. It’s just how you view it. Since it’s work assigned to me, I’ll do my best. That’s how I think…*


#### 3.8.2. Opportunity for a New Experience

The participants had a positive attitude toward gaining work experience related to a novel infectious disease, which regular nurses do not experience. They also gave added meaning to their experiences by viewing them as precious time for learning the skills or work competencies of veteran nurses.


*Caring for COVID-19 patients… It’s not something you can do just because you want to. I experienced infectious disease patient care, which other nurses could not. I am thankful.*

*Because there were nurses from various wards, many nurses had much more experience than me. I learned a lot from watching those nurses carry on with their work. I thought that my competencies could be upgraded a level…*


#### 3.8.3. Pride as a Committed COVID-19 Nurse

Most participants expressed satisfaction in knowing that they are making a specific contribution to the fight against a global pandemic. Their pride increased even further with the heightened social recognition for nurses who cared for COVID-19 patients.


*The work that I’m doing is truly helping someone else. I am contributing during this national disaster situation. I am there at this historical moment…*

*In the beginning, there were a lot of broadcasts with touching stories about the occupation of nursing. The feeling of others looking at us differently? Because of being COVID-19 hospital nurses… there is a sense that everyone is looking at us somewhat differently.*


### 3.9. Taking Another Step in One’s Growth

#### 3.9.1. Providing Real Nursing

In the early stage of caring for COVID-19 patients, the participants struggled with unfamiliar work and, at times, experienced self-doubt. As time passed, however, they personally completed work that they used to delegate to nursing assistants and shifted to the point where they were examining and being concerned about the well-being of the patients.


*I didn’t need to feed them since nursing assistants or family caregivers normally help them with their meals. However, I double checked to make sure there is no risk of aspiration while being fed. Since I’ve been changing their positions, I think about what could be done to reduce bedsores…*

*Before, I didn’t think I looked after the patient’s mood. But now, those things come into view. You notice their changes in expression, and the voice of the patient when his or her neighbor in the next bed is discharged. They feel depressed. That’s why we came up with the idea of getting song requests from the patients and playing those songs. Young patients even dance inside the room and really liked it.*


#### 3.9.2. Broadening Perspectives

The participants expressed that as time went on, they became more familiar with caring for patients in the isolation unit and that they could see the entire isolation unit at a single glance. Among the participants, some even proposed implementing counseling programs for patients and nurses, systematic infection control guidelines, as well as increasing staffing for successful infection control, and providing high-quality medical service while the pandemic lingers on.


*We need to train new nurses on donning and doffing PPE like in real-life situations, and we also need stress and depression management for nurses, since caring for COVID-19 patients is becoming long-term.*

*Additional staffing is needed the most. We also need symptom assessment by telephone to minimize contact with the patients and a system to enable drug administration and treatments to take place according to mealtimes.*


#### 3.9.3. Confidence in Caring for Infected Patients

In the early stage of dedicated COVID-19 nursing work, most participants had fears and worries about the novel virus. However, as they spent months on the frontline fighting against the infectious disease, they gained confidence in their nursing work, from wearing PPE to the overall nursing skills required to care for infected patients. Among the participants, there were some nurses with patient care experience during the MERS epidemic. However, regardless of whether they had previous experience, most participants stated that they would be much more competent when another novel infectious disease outbreak occurs.


*Now, there is a lot less fear. After caring for infected patients for seven months, I believe I’ve gained a certain level of competency in caring for infected patients, and I can do the job when such an infectious disease outbreak occurs again.*

*It’s not like Wonder Woman or Superman, but the thought that I can be helpful when another infectious disease outbreak occurs since I’ve gained experience in caring for infected patients. I also have the feeling of wanting to experience it.*


## 4. Discussion

The first theme described by the nurses’ COVID-19 patient care experience was Being Pushed into the Battlefield Without Any Preparation. As the number of confirmed COVID-19 cases increased rapidly in South Korea, the participants’ workplace was designated as a COVID-19 hospital; thus, they rapidly prepared to accept patients and provide patient care to those diagnosed with COVID-19, while experiencing fear from an unfamiliar pandemic.

This situation was completely different from the results described in a previous study [18] that reported participants who “consented” to “join” others in COVID-19 patient care and those in another study [25] that reported on nurses who volunteered to provide MERS patient care as they had grown tired of their existing work and wanted a new challenge. The participants in this study were obligated to provide COVID-19 patient care due to their job, regardless of their preference, and experienced high levels of stress due to being unprepared. Previous research has described the increased mental health challenges experienced by healthcare professionals in COVID-19 hospitals compared to those who volunteer to work in COVID-19 patient care [17]; thus, identifying policies that can enhance the mental health of COVID-19 hospital nurses is needed.

Moreover, most participants stated that they were very anxious during the early stages of the pandemic due to their assignment in a COVID-19 isolation ward without systematic and sufficient training for PPE donning and doffing, consistent with previous findings [20]. On the other hand, nurses who had experience with direct patient care during the MERS epidemic expressed relatively less anxiety about becoming infected than those without MERS experience. A similar context was reported in a previous study [19] which found that nurses with previous infectious disease nursing experience had relatively less anxiety than nurses who experienced COVID-19 patient care without comparable infectious disease experience. The participants believed that they knew about PPE donning and doffing to some degree based on participating in contests or video training at the hospital following the MERS epidemic, but the real-world experience was entirely different. Putting PPE on (donning) appropriately can reduce respiratory infections, but self-contamination often occurs during the incorrect removal of PPE (doffing) [26,27,28]. Therefore, if the nurses fail to follow the correct procedures due to insufficient education and training related to PPE donning and doffing, both the nurse and the patient can face an increased risk of infection [29]. This is a very critical issue in terms of the safety of nurses and patients. Therefore, proper education and training related to PPE donning and doffing is needed. Aliakbar et al. [26] reported that nurses with high levels of knowledge about infection control but below-average practice needed further practice and nurse training programs to address such discrepancies. Therefore, practical training on PPE donning and doffing, new nurse training that includes actual content about infection guidelines, and regularly scheduled on-the-job training are needed in clinical settings to reduce nurses’ anxiety about infection and increase their likelihood of properly coping with the situation.

The second identified theme was Struggling on the Frontline. Amidst the unfamiliar and unpredictable chaos of the pandemic, the participants experienced various stressors while caring for COVID-19 patients. Confusion regarding the frequent changes in infection control protocol has also been reported by nurses during other global pandemic situations, including MERS [25] and human swine influenza [30]. The nonspecific and frequently changing infection control guidelines not only resulted in confusion, but also caused nurses to experience anxiety and stress due to decreased trust [31]. When caring for infected patients, nurses’ confidence in their ability to protect themselves from infection is positively correlated with their willingness to provide patient care [32], suggesting that infection control guidelines must be specific and clear to increase nurses’ confidence in their ability to avoid infection.

In addition, the nurses experienced challenges from wearing PPE, with increased discomfort as the duration of time they wore PPE increased. This difficulty is commonly reported in qualitative studies of infectious disease patient care [18,19,20,25,33]. Chinese nurses who worked over eight hours while wearing PPE and adult diapers while providing COVID-19 patient care expressed that they felt like collapsing [19]. However, the participants in the present study mentioned that the conditions were bearable in the early stage of the pandemic as they spent relatively short periods in the isolation unit while wearing PPE due to sufficient nurse staffing. Sufficient nurse staffing allowed the nurses to be divided into several teams during each shift and enter the isolation units in turns. However, as nurse staffing gradually faced shortage, the participants felt burnt out as the duration of time wearing PPE increased. Therefore, while developing PPE that can reduce nurses’ discomfort is needed, a strategy to reduce the duration of time wearing PPE by reducing the time nurses spend in isolation units by increasing staffing levels is also necessary.

The participants experienced a sense of shame from completing tasks that were not usually their responsibility, such as waste management, hospital room cleaning, and facility inspection. Another study [25] reported a similar context, in which the participants felt it was unfair that they were required to perform room cleaning and disinfection. These responsibilities were a source of stress resulting from the nurses being responsible for everything in the isolation unit by minimizing the number of people in contact with infected patients. Therefore, strictly implementing training on PPE donning/doffing and infection control for medical assistants and establishing policies that allow nurses to focus on patient care through the division of work should be required.

The third theme was Altered Daily Life. The participants were fearful that their contact with COVID-19 patients would result in the unintended infection of others, especially their own family. They regularly traveled between their home and the hospital and practiced strict social distancing while having anxiety about possibly being infected whenever they experienced any change in their health, consistent with previous studies [18,19,34]. When a colleague in the hospital became infected, the level of anxiety in the participants increased as it could have also happened to them. After the nurse became infected, hospitals began to screen staff once a month, which is a hospital policy not previously reported. Therefore, regularly scheduled screening is necessary to reduce anxiety about potential infection among nurses who provide infectious disease patient care and for the safety of nurses and patients.

The fourth component was Low Morale. The participants felt that nurses did not receive the respect they deserved despite experiencing a high level of physical and psychological stress from their contact with COVID-19 patients. As specific compensation guidelines have not been determined, the participants had not received any financial compensation, but reported low morale after hearing rumors that even if they were compensated, it would be less than that received by the doctors. When establishing policies in preparation for infectious disease outbreaks, it is necessary to establish compensation guidelines for healthcare professionals who work closely with infected patients. In a study where the hospital paid for anti-epidemic health insurance [19], and the government and hospital provided financial compensation and issued awards for nurses’ contributions [33], the nurses reported that the value of their work was recognized and expressed that they were “being supported” by their organization. As the hospital that the participants are affiliated with is highly likely to be designated again as an infectious disease hospital during another novel infectious disease outbreak, establishing compensation-related guidelines must be developed to lift the morale of nurses.

Meanwhile, the nurses experienced stigma because they provided care to COVID-19 patients. COVID-19 spread rapidly to many more people than previous infectious diseases, leading to the stigmatization of COVID-19 patients and healthcare professionals who treat them [33]. The stigma described by the participants was similar to that in a study reporting that nurses caring for MERS patients were forbidden from riding in elevators, and their children’s kindergarten attendance was denied [25,35]. Moreover, when a nurse colleague became infected, the public criticized the nurse based on an unfairly strict infection control standards, resulting in the participants feeling anger and skepticism. These experiences contributed to low morale from thoughts, such as “Why do we have to sacrifice for the patients and be criticized?”

No study has reported on the effect of stigma experienced by nurses who cared for COVID-19 patients on post-traumatic growth, but findings from previous research about stigma and post-traumatic growth of adults with HIV [36,37] are noteworthy. The stigma experienced by adults with HIV had a negative effect on post-traumatic growth [37]. Similar to what was shown in the present study, adults with HIV experienced stigma, which affected their patterns of disclosure and eventually prevented them from disclosing their trauma experience (HIV-related challenges) [36]. However, self-disclosure is a vital factor in coping because it is the first strategy for managing stress from a trauma experience and the psychological preparation for post-traumatic growth [38]. The higher the level of self-disclosure, the higher the level of post-traumatic growth, and if stigma prevents self-disclosure, post-traumatic growth can also be suppressed. Such stigma causes the nurses caring for infected patients to feel increased stress and had a negative effect on their mental health [39,40,41]; thus, countermeasures to reduce social stigma are needed. South Korea implemented strict prevention and control policies in the initial stage of the pandemic and disclosed information about the routes of movement for infected persons, including what they did and with whom. As a result, the infection nurse was criticized for having a meal with her parents. Based on our findings, identifying methods of balancing strict infection transmission prevention policies and protecting personal privacy is necessary. Moreover, increased public awareness about healthcare professionals becoming infected not due to non-compliance with prevention and control rules, but the nature of their work, which requires close contact with patients infected with a high-risk virus, would be beneficial.

The fifth component was an Unexpectedly Long War. The participants expected the pandemic to be resolved within a short period, similar to the MERS epidemic. However, they felt helpless when faced with not knowing when the pandemic would end and experienced burnout from decreased staffing for COVID-19 patient care as a result of admitting regular patients to some wards following financial problems faced by the hospital.

As the time spent in the isolation unit doubled due to staffing shortages, the participants experienced extreme stress and fatigue. The number of patients kept rapidly increasing, and the shortage of nurses caused tremendous fatigue, comparable to the situation described in a study from China in which some staff were reported to be sleeping while standing due to extreme fatigue [19,33].

Burnout in nurses negatively affects their physical and mental health, can contribute to a decline in patients’ quality of care, and potentially threatens patient safety [42]. Therefore, strategies to prevent burnout among nurses during an epidemic are needed. Research has indicated that 90.5% of frontline healthcare professionals who treat COVID-19 patients experience burnout [42], and these rates may increase as weekly working hours become longer and job stress becomes higher [10,13], making additional staffing critical for reducing burnout. A sample of nurses who cared for MERS patients did not receive sufficient rest nor their requested vacation time of 1–2 weeks [25]. Similarly, the current participants expressed that they needed a long vacation. Employee welfare policies that provide vacation time and reduce work hours are needed. Specifically, dedicated infectious disease hospitals must have governmental financial support as they need to have high staffing levels in the wards with infected patients, rather than admitting regular patients, even when the number of infected patients decreases.

Moreover, the participants could not perform their regular nursing duties for an extended period, resulting in concerns about their own nursing competency when they eventually return to regular nursing duties. This has not been previously reported, which could be attributed to other comparable infectious disease epidemics being shorter than the COVID-19 pandemic and other COVID-19 studies having been conducted during the early stage of the pandemic; thus, problems associated with its protracted length have not been identified. This finding is particularly meaningful as nurses with limited experience may be faced with losing the opportunity to learn much about patient care, suggesting that devising measures to deal with a prolonged infectious disease outbreak would be beneficial. In other words, policies are needed to allow nurses with limited experience to quickly adapt to their duties through re-education and training on patient care in the general ward before they return to their general nursing duties after the pandemic ends.

The sixth component was Ambivalence Toward Patients. With respect to the characteristics of COVID-19 patients, which differ from regular patients, the nurse participants described both positive and negative feelings toward the patients. In South Korea, cluster outbreaks of COVID-19 involving specific religious and anti-government protest groups that were not following infection control guidelines occurred. As a result, the participants reported a negative bias toward some patients, which is a novel finding for this study. The negative emotions, stigma, and discrimination that nurses had toward patients could influence the care provided to the patients [41], suggesting that strategies for reducing stigma toward patients are needed. Therefore, developing intervention programs that help nurses recognize negative attitudes and enable nurses to recognize any negative biases toward patients could result in patient stigmatization, just as nurses were stigmatized for providing patient care to those with an infectious disease.

Moreover, the participants described feelings of anger when patients refused to accept that they were infected with COVID-19 and would not cooperate with treatment, or when patients made trivial requests of the nurses. Such findings have been reported previously [25] in which MERS-infected patients projected their anxiety, fear, and stress on nurses during their quarantine and showed uncooperative attitudes. Therefore, education programs on the emotional state of patients admitted to the isolation unit should be implemented to allow the nurses to understand patients’ emotions and behaviors. Inpatient education that increases patient awareness regarding the challenges of infectious disease patient care, and its difficulty compared to caring for regular patients and potential burnout, and that provides information about behaviors that are not permitted inside the hospital should be conducted. However, nurses reported positive feelings toward the patients, as was previously described in one study showing that the nurses sympathized with patients and that the emotional distance between patients and nurses was reduced [25].

The seventh component was the Forces That Keep Me Going. Previous studies have reported that nurses were grateful to those who supported them [18,19], moved by people referring to them online as heroes and sending relief supplies [19,33], and gained strength from letters written and sent by children [25].

All participants stated that the strongest force to keep them going was their workplace community, which has been previously described [25]. In a previous study of military personnel, resilience was found to have a larger influence on team cohesion [43], while peer support was found to serve as a protective factor for burnout among emergency room nurses during the MERS epidemic [44]. When experiencing high stress along with a colleague, peer support could be a powerful resource; thus, strategies for strengthening peer support could be beneficial. The participants mentioned that they became closer with colleagues by talking with each other during their breaks after working in isolation units with infectious disease patients. Therefore, providing sufficient break time for nurses who care for infected patients and allowing them to have their own time and space for breaks is necessary.

The eighth component was Giving Meaning to My Work. The participants gave new meaning to their work and discovered new values in this difficult situation. Such findings are supported by other studies that reported nurses find meaning in their own experiences, including viewing their infectious disease patient care experiences as unique and gift-like [19], as a precious, meaningful experience that helps their work experience [33], as being lucky to have the opportunity to fight against the virus [18], or as being a proud experience as a nurse who provided high-quality care during a national disaster [25].

Moreover, the participants felt proud and had a sense of duty from hearing those around them, acknowledging that “nursing is difficult, but is admirable work”. Instead of having numerous broadcasts on the challenging work of nurses during the early stage of an infectious disease epidemic, it is necessary to provide nurses with continued support to help feel that their work has value beyond just being a job.

The ninth component was Taking Another Step in One’s Growth. The participants could provide higher quality physical and psychological care than before and gained confidence in their infectious disease patient care over time while developing a broader perspective than at the early stage of the pandemic. In hospitals in South Korea, assisting with meals, changing diapers, and changing positions in the general ward, not ICU, are performed by family members or caregivers, not nurses. The participants reported feelings of self-doubt from having to perform such work that was unfamiliar to them, but as time went on, they became more concerned with the well-being of the patients and contemplated how to provide high-quality care. Moreover, they were interested in the depression and sense of isolation felt by the patients in the isolation unit and provided emotional care by consoling them. Some participants stated that they were not familiar with providing emotional care. When patients with infectious disease are admitted to the isolation unit, they experience various emotional problems, including anxiety, depression, and aggressiveness [44,45,46]. Therefore, education on both infectious disease control methods and emotional changes in infected patients, and the emotional care for such patients, is needed. In the beginning, the participants were busy taking care of whatever appeared in front of their eyes, but as time passed, they thought about what they could propose to improve the infectious disease control system. Proposals derived from their experiences could serve as basic data for establishing infection control systems and policies during a global pandemic.

In the present study, the experiences of COVID-19 hospital nurses who provided care to COVID-19 patients involved struggles on the frontline of a global pandemic, but they eventually grew as they discovered the value and meaning to their work while receiving social support from their community. This process is very similar to post-traumatic growth, meaning that their growth exceeded the level prior to a painful experience by reflecting on the meaning of the experience and using various resources following a life-altering painful experience [38]. From the perspective of post-traumatic growth, the participants experienced post-traumatic growth through a “deliberate rumination” process of seeking value and giving meaning to their own experience while receiving support from friends, colleagues, and the public. Their experiences involving gaining confidence about infectious disease patient care by successfully performing work that previously they were not confident about, feeling grateful toward those who supported them, forming deeper relationships with patients, and taking on work with a greater sense of duty than before. These experiences encompass all three domains of post-traumatic growth: “a changed sense of self”, “a changed sense of relationships with others”, and “a changed philosophy of life” [47].

As examined above, while the outbreak of a novel infectious disease cannot be prevented, healthcare professionals, especially nurses, gain experience and influence as they provide care for patients on the frontline. Providing material compensation and psychosocial support at a personal level, as well as improving the nursing environment by supplementing infectious disease control systems at organizational and social levels and establishing relevant policies, is necessary to reduce nurses’ negative experiences and promote positive experiences, such as post-traumatic growth.

Among existing research, most studies have suggested the need for institutional improvement that focuses on the negative aspects of nurses’ experiences during a pandemic. However, the current study comprehensively explored the lived experiences of nurses who were caring for COVID-19 patients during a pandemic and was focused on post-traumatic growth, which can be viewed as a positive outcome. Studies that focus on the post-traumatic growth of nurses caring for patients with infectious diseases could shift existing research paradigms during a global pandemic. While preventing future outbreaks of infectious diseases is difficult, the present findings are meaningful in that they provide basic data that could be used to establish and refine hospital systems and policies that support nurses on the frontline of infectious disease control that would allow them to adapt quickly and have positive experiences.

Furthermore, the themes and subthemes that we found in this paper could prove useful to researchers around the world as they investigate the nuances of the pandemic in their own countries. Because we were able to study nurses’ experiences of the pandemic, our research can provide a baseline for those in other countries who may be experiencing the pandemic in different ways based on differing cultural backgrounds and experiences with COVID-19. The themes of experiences of COVID-19-designated hospital nurses may be universal, but they may play out differently depending on context. Thus, our research can provide a starting point for global research as the pandemic unfolds throughout the world. This could help to explain not just this specific pandemic in this exact location, but how crises are managed in different ways by different populations.

In addition, the participants expressed a wide variety of experiences related to COVID-19 patient nursing, and we discussed their main experiences with a focus on the post-traumatic growth process. However, we were also able to identify a change in their professional role of being a nurse because of their experience. Therefore, it would also be meaningful to focus on the professional role change process that emerges from the experience of nurses taking care of patients with novel infectious diseases such as COVID-19 and compare it with professional role theory such as Benner’s [48] model.

### Limitations

The use of convenience sampling in the current study is a notable limitation. The participants in the present study were nurses working in a COVID-19-designated hospital located in the second-largest city in South Korea (i.e., Busan). Further, the participants did not volunteer to provide COVID-19 patient care. Moreover, the hospital uses a system that transfers critically ill patients to regional tertiary hospitals; thus, the participants had extremely limited nursing experience caring for critically ill COVID-19 patients. Therefore, caution must be exercised in attempts to generalize these results. Additional studies are needed that include other populations of nurses who can share diverse experiences, such as nurses who volunteered as frontline workers who worked with COVID-19 patients and those who provided care for critically ill COVID-19 patients. Furthermore, the data collection began about seven months after the onset of the COVID-19 pandemic. Some of the participants discussed their seven-month patient care experience, but a few of them had a relatively short period of nursing in this situation, about two months. There may be differences between the experiences of the individual participants in this study and the essence of their COVID-19 patient care experiences at this point in time, nearly a year after the onset of COVID-19 pandemic.

## 5. Conclusions

Nurses’ experiences while caring for COVID-19 patients that were identified in the present study suggested, “growth through struggling on the frontline against a global pandemic”. The participants provided care to COVID-19 patients without any preparation, which resulted in a number of challenges on the frontline. However, they progressed toward growth by enduring these difficulties using social support and giving new meaning to their work. Epidemics will occur in the future, making it necessary to develop systematic infection control education and appropriate job manuals that reflect actual practice and devise educational programs that could enhance the physical and mental health of patients and nurses. Moreover, policy studies that established methods of ensuring compensation for the difficult work performed by nurses and policies on staff allocation and work type during infectious disease outbreaks are recommended. Given the current findings, nurses who provide care to patients with an infectious disease have both negative and positive experiences, including post-traumatic growth; therefore, exploratory studies that examine factors associated with nurses’ post-traumatic growth in this context are needed to develop post-traumatic growth programs for such nurses. It is critical to realize that public health policies in a pandemic situation must focus not only on patients, but also on frontline healthcare workers, including nurses.

## Figures and Tables

**Figure 1 ijerph-17-09015-f001:**
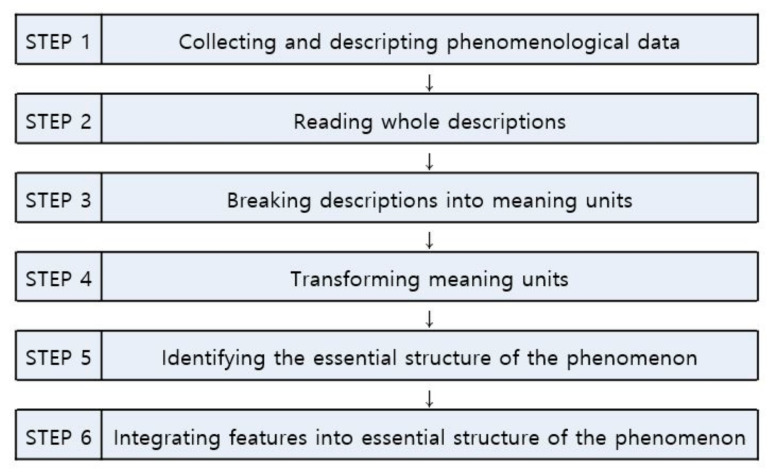
Concrete steps in Giorgi’s phenomenological method.

**Table 1 ijerph-17-09015-t001:** Themes and Subthemes identified from the lived experiences of nurse participants providing care to COVID-19 patients.

Themes	Subthemes
Pushed onto the Battlefield without Any Preparation	Unavoidable Duty
	Limited Preparation Time
	Fear of an Unknown Enemy
Struggling on the Frontline	Constantly Changing Guidelines that Lack Details
	Discomfort Due to Personal Protective Equipment
	Unfamiliar Work
	A Series of Unexpected Situations
Altered Daily Life	Reinforced Infection Policies Within the Hospital
	Becoming Sensitive to Even Minor Symptoms
	Voluntarily Restricting Social Activities
	Cohort Isolation after the Confirmed Diagnosis of a Colleague
Low Morale	My Labor Is Not Being Properly Recognized
	Being Treated Like a Virus
	Strict Social Standards
Unexpectedly Long War	Despair with No End in Sight
	Tired Body and Mind
	Concerns about General Nursing Competency
Ambivalence Toward Patients	Having a Bias
	Becoming Angry at Uncooperative Attitudes
	Feelings of Pity
	Solidarity with the Patients
Forces That Keep Me Going	Concerns from Family and Friends
	Patients Showing Their Appreciation
	Public Support
	Strengthened Camaraderie
Giving Meaning to My Work	Called to Do the Work Expected of Me
	Opportunity for a New Experience
	Pride as a Committed COVID-19 Nurse
Taking Another Step in One’s Growth	Providing Real Nursing
	Broadening Perspectives
	Confidence in Caring for Infected Patients

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
