# Peer review of "South Korean Nurses’ Experiences with Patient Care at a COVID-19-Designated Hospital: Growth after the Frontline Battle against an Infectious Disease Pandemic"

_ijerph, 2020, doi:10.3390/ijerph17239015_

Round 1
Reviewer 1 Report
I admire the effort developed to conduct the study. I understand the limitations and context. I have some comments related with the manuscript. The results and conclusions are very interesting.
1. Giorgi´s method explanation is not enough (lines 84-90). I suggest to include a short diagram about how the method works and the information processing.
2. In the conclusion section, Try to include some recommendations about how the list of themes and subthemes can be used in other countries to explore differences and coincidences with the present study.
Author Response
We thank you for your thoughtful suggestions and insights.
The manuscript has benefited from these insightful suggestions.
Please see the attachment files.

Reviewer 2 Report
Thank you for the opportunity to review your work. Overall your manuscript was well written and provided insight into nurses experiences with caring for individuals suffering from COVID-19. Below are a few suggestions to improve the clarity of your manuscript.
- Within section 2.1 while you provide a brief overview of Giorgi's method a bit more detail in this section would provide much needed detail for readers not familiar with this method. Providing a bit more information on how this method was specifically used in the study would be helpful.
- Within section 2.3 was a semi-structured interview guide utilized for the interviews? If so this is not clear and including this information as well as specific questions asked during the interviews would add clarity to this section.
- Within section 2.4 it is unclear who participated in the meetings described. Was this the research team or did the participants also participate in these meetings?
- While figure 1 is a nice visualization of the identified themes I'm not sure it adds value to the manuscript.
- Within the discussion the authors describe the participants concerns related to insufficient knowledge related to donning and doffing PPE but there is no connection to the wealth of evidence related to infection control and patient safety/quality. This evidence should be incorporated to strengthen the manuscript.
- Further within the discussion there Stigma is discussed and it would be interesting to link any existing evidence from the AIDS pandemic to determine if pertinent studies are available to enrich this section of the discussion.
- Within the limitations section there is no mention of what appears to be a homogenous sample which also further limits generalizability. This should be reflected within the limitations.
Author Response

(The authors gave the same response as above.)

Reviewer 3 Report
Thank you for the opportunity to read this manuscript. The manuscript is very interesting and deals with a highly relevant topic. The introduction and method is well written and only requires a few minor revisions. My main concern is the findings that in my opinion is a bit difficult to grasp and sometimes also underdeveloped. I actually do not see how the chosen analysis method (phenomenology) is reflected in the findings. Parts of the findings are more of a description of the situation than on the nurses lived experiences of the situation. I believe that the manuscript would benefit from a deeper analysis of the actual lived experience. The lived experiences is more outspoken in the discussion and can be further developed in the result section as well.
Introduction
The introduction is well-written and easy to understand. Relevant aspects are brought up and in my opinion I do not see anything that needs to be revised. Still, consider if you want to update the nubers of confirmed Covid-19 cases and deaths as these number unfortunately are rising in a high speed right now.
In relation to your findings I think you have to reconsider your aim. This comment is thorougly described under Findings.
Method
The data analysis section needs to be further developed. At this point it is hard to really understand how the analysis have been conducted. For example, when was the meaning units separated from the collected data and how was the meaning units identified? Does the specific analytical procedure that is described include all data or just the selected meaning units? How did you handled your own preunderstanding? Please, expand the description of your data analysis procedure in order to make it more transparent for the reader.
Regarding dependability it is stated that you compared the derived meanings – how did you compare? Please, expand this text in order to clarify your procedure.
Results
The results include some very interesting and important findings, however it also includes some challenges. Phenomenological studies aim to describe the lived experience of the participants and that is sometimes done in this manuscript. Still, I see that the results also includes a lot of text that are more of a description of the actual situation rather than the lived experiences. For example in 3.1.2, 3.2.2 and 3.2.4.
I also see that some of the subthemes are related to each other but described separately. For example there are several subtemes that I interpret as being related to the lived experience of a changing professional role of being a nurse (3.5.3, 3.4.1, 3.2.3, 3.2.4, 3.6.2 and 3.8.2).
Not until the last paragraph of the results I really understand that you are trying to present your results in a longitudinal matter. Such information in the beginning of the results would have made it easier to read and understand your results. Please consider to present the essential structure and figure 1 earlier in the text.
If you present 3.10 and figure 1 earlier in the text and go through your results once again in order to clarify and highlight the lived experiences even more I believe that the quality of your resuls will improve. Also consider to merge some of the subthemes.
If you want to keep the descriptions of the work situations without relating those to the lived experiences of the participants you might consider the choice of analysis method.
Discussion
As the discussion narrowly mirrors the themes and subthemes in the result section it also tends to be shattered. This makes it sometimes hard to follow your reasoning and I think that the quality of the manuscript would improve if some of the subthemes were merged in the results and also in tín the discussion section. Still, the discussion is very well written and the lived experiences of the participants are more appearent here than in the result section.
In the end of the discussion you are referring to post-traumatic growth and relate your findings to this phenomena in a really interesting way. This paragraph also highlights the longitudinal aspect of your work which is not that apparent at the moment. I suggest that you clarify how you have structured you findings in order to increase the readability. However, if you choose this revision you also need to mention the possible limitations with that choise. As I understand it the interviews have been conducted in varius time points and some nurses have experiences from several months while others have experiences from just a few. This needs to be brought up in 4.1 Limitations.
Author Response

(The authors gave the same response as above.)

Round 2
Reviewer 3 Report
Thank you for the revisions made. I think that the quality of the manuscript has improved and is now suitable for publication.